# The Effect of Waste Plant Oil on the Composition and Micro-Morphological Properties of Old Asphalt Composition

**Zhi Suo** [1,2,3,*] **, Lei Nie** [1] **, Fanrong Xiang** [1] **and Xu Bao** [1]

1   School of Civil and Transportation Engineering, Beijing University of Civil Engineering and Architecture, Beijing 100044, China; buceaer@163.com (L.N.); x_x20210101@163.com (F.X.); baoxu1996@163.com (X.B.)
2   Beijing Advanced Innovation Center for Future Urban Design, Beijing University of Civil Engineering and Architecture, Beijing 100044, China
3   Beijing Urban Transportation Infrastructure Engineering Technology Research Center, Beijing University of Civil Engineering and Architecture, Beijing 100044, China
*   Correspondence: suozhi@bucea.edu.cn

**Abstract:** At present, the regeneration agent with mineral oil as the main component is widely used. However, its strong volatility, nonrenewability, and other shortcomings restrict the development of regeneration technology. In this study, waste vegetable oil was used as the main component of the regeneration agent to prepare regenerated aging asphalt. The change in microstructure of aging asphalt was explored with the change in waste vegetable oil content and regeneration time. The results showed that the addition of waste vegetable oil could effectively supplement the contents of saturates and aromatics, and inhibit the oxidation of saturates and aromatics into asphaltene and resin. When the regeneration time was 60 min and the regeneration content was 7.0%, the effect was best. The correlation between the component composition of regenerated aging asphalt and the micro-morphological characteristics was in the order of saturates, asphaltene, resin, and aromatics.

**Keywords:** waste vegetable oil; aging asphalt; asphalt component; micro-morphology characteristics





## 1. Introduction

The aging phenomenon of asphalt material is affected by ultraviolet radiation, temperature, rain, and other factors during the use of asphalt pavement. Due to environmental factors such as alternate temperature changes in the four seasons, the micro-morphology of the asphalt material will also change, resulting in obvious changes in the macroscopic performance indexes such as the softening point, ductility, and viscosity of asphalt during aging, and shortening the service life of the pavement [1]. Asphalt is a very complex chemical material. The changes in its four components, content, molecular weight, functional groups, and other chemical compositions have a fundamental impact on asphalt pavement performance [2]. The esters of straight-chain higher fatty acids and glycerol is not volatile at high temperature. Therefore, they can be used to recover the micro-chemical composition of bitumen and the change in colloidal stability of asphaltene [3].

Qu [4] and Menapace et al. [5] separated matrix asphalt with different aging degrees of the Corbett separation method. The experiment concluded that with the decrease in aromatic content of asphalt in short-term aging, the asphaltene content increased. The chemical reaction occurred after the aging of matrix asphalt, and part of the aromatic content and resin became asphaltene. Xing et al. [6] analyzed the changes in functional groups of matrix asphalt before and after aging by atomic force microscopy-infrared spectroscopy, and concluded that with the increase in aging, the content of carbonyl and sulfoxide increased.

Exploring the changes in asphalt performance before and after aging mainly serves to improve the regeneration ability of aging asphalt. The waste edible oil regeneration agent [7], waste oil regeneration agent [8], and waste bio-oil regeneration agent [9] are

three kinds of products with great transformation potentials to realize large-scale industrial production and practical application. Among them, the main raw materials of the waste edible oil regeneration agent are recycled waste cooking oil [10] and expanded food production waste oil as the representative, and the main raw materials of the waste biological oil regeneration agent are waste tung oil [11] and cotton seed oil [12] as the representative. The main raw materials of the waste oil regeneration agent are motor vehicle waste oil [13] and waste hydraulic oil [14]. These oils are rich in a variety of small-molecule organic compounds, which can be mixed with recycled asphalt to compensate for the lack of light components caused by the aging of asphalt to achieve regeneration.

The effects of the vegetable oil regeneration agent on the performance of aging asphalt in terms of the physical properties, the rheological properties, and the road performance of aging asphalt were analyzed by Leng [15], Hugener [16], and Chen [17] compared with waste vegetable oil and mineral oil. The results showed that the regeneration effect of the former is better than that of the latter, in terms of the regeneration degree of aging asphalt regenerated by the vegetable oil regeneration agent. Somé [18] compared the amount of aging asphalt regenerated by the vegetable oil regeneration agent and mineral oil regeneration agent to the same level, and they found that the amount of mineral oil regeneration agent was three times that of the vegetable oil regeneration agent.

Based on the above research results at home and abroad, waste vegetable oil was used as a regeneration agent for regenerated wet aging asphalt at room temperature. There is little research on the influence and correlation of waste vegetable oil as a regeneration agent on the composition and micro-morphology characteristics of aging asphalt through laboratory experiments. The research results have positive significance for the development of cold regeneration for aging asphalt.

## 2. Materials and Methods

Waste vegetable oil was used as a regeneration agent to regenerate aging asphalt. The changes in microstructure of regenerated aging asphalt were studied by atomic force microscopy and Fourier-transform infrared spectroscopy, and the relationship between microscopic changes was analyzed.

### 2.1. Preparation of Regenerated Aging Asphalt

2.1.1. Preparation of Waste Vegetable Oil Regeneration Agent

In the early stage of our research group, three kinds of vegetable oils, sunflower, soybean, and corn, were simulated on aging to obtain waste vegetable oils. After aging, they had the same functional groups and could be used as regeneration agents [19]. Through market research, soybean oil was most representative. Therefore, waste vegetable oil after cooking was simulated by heating soybean oil in an oven at 180 °C for 6 h. The performance indexes are shown in Table 1.

**Table 1.** Performance index of waste vegetable oil.

| Technical Index | Unit | Measured Value | Recommended Value |
|---|---|---|---|
| Density | g/cm$^3$ | 0.9 | Actual measurement |
| Viscosity | Pa·s | $1.2 \times 10^{-1}$ | 0.1–20 |
| Surface tension | 10-3 N/m | 46 | $\geq$36 |

2.1.2. Extraction of Aging Asphalt

The extraction of aging asphalt from the waste asphalt mixture is one of the effective means to accurately evaluate the aging performance of aging asphalt. In this study, the commonly used centrifugal separation method was used to recover the aging asphalt. According to the standard of test procedures for asphalt and the asphalt mixture in China, the asphalt solution was recovered, and the aging asphalt was purified by the rotary evaporator method. Recovered aging asphalt was tested for three indicators based on viscosity. The test results are shown in Table 2.

**Table 2.** Performance index of aging asphalt after extraction.

| Technical Index | Unit | Measured Value | Test Method |
|---|---|---|---|
| Penetration degree | °C | 16.3 | T0604 |
| Softening point | 0.1 mm | 78.8 | T0605 |
| Ductility | mm | 86 | T0606 |
| Viscosity | Pa·s | $9.9 \times 10^{-1}$ | T0625 |

2.1.3. Laboratory Simulation of the Preparation of Aging Asphalt

Through market research, Xinhai-70 # asphalt, which is frequently used in this area, was selected to prepare aging asphalt. According to the standard of test procedures for asphalt and asphalt mixtures in China, its basic performance test meets the requirements. The results are shown in Table 3.

**Table 3.** Basic performance index of the original asphalt.

| Technical Index | Unit | Measured Value | Recommended Value | Test Method |
|---|---|---|---|---|
| Penetration degree | °C | 70.1 | 60–80 | T0605 |
| Softening point | 0.1 mm | 49.7 | ≥43 | T0604 |
| Ductility | mm | 207 | ≥100 | T0606 |
| Viscosity | Pa·s | $3.8 \times 10^{-1}$ | — | T0625 |
| Wax content | % | 1.4 | ≤2.2 | T0615 |
| Flash point | °C | 299 | ≥260 | T0611 |

Experimental data to see whether the performance indicators of Xinhai-70# asphalt meet the requirements.

The aging asphalt A# was prepared by heating Xinhai-70# asphalt to 165 °C for 75 min by a rolling thin-film oven test (RTFOT). After the preparation of aging asphalt A#, aging asphalt B# was prepared by the pressure heating aging method (PAV) at 110 °C, 2.1 MPa, and 20 h, and aging asphalt C# was prepared by the pressure heating aging method (110 °C, 2.1 MPa, and 40 h). Three kinds of aging asphalt A#, B#, and C# were obtained. The basic properties of the three aging asphalt types were tested. The test results are shown in Table 4.

**Table 4.** Simulation of the basic performance index of aging asphalt.

| Technical Index | Unit | A# | B# | C# | Test Method |
|---|---|---|---|---|---|
| Penetration degree | °C | 41.6 | 30.4 | 18.9 | T0604 |
| Softening point | 0.1 mm | 45.5 | 49.6 | 55.2 | T0605 |
| Ductility | mm | 206 | 321 | 107 | T0606 |
| Viscosity | Pa·s | $6.0 \times 10^{-1}$ | $7.2 \times 10^{-1}$ | $8.6 \times 10^{-1}$ | T0625 |

The basic properties of aging asphalt after extraction and the simulation of the three kinds of aging asphalt, A#, B#, and C#, were compared. The basic properties of aging asphalt C# and aging asphalt after extraction were the most similar. Therefore, aging asphalt C# was selected as a comparative sample of the following studies.

At room temperature, the waste vegetable oil was used as a regeneration agent to regenerate aging asphalt C#, which was fully fused by standing. The aging asphalt with different regeneration degrees was obtained by different dosages of the regeneration agent and different regeneration times.

(1) Determination of different dosages of the waste vegetable oil regeneration agent

According to the previous research of the research group, it has been concluded that when the vegetable oil content is 6.0%, the penetration, softening point, ductility, and other performance indexes of the aging asphalt can be restored to the state of the original asphalt. Therefore, it is determined that the dosage of waste vegetable oil is based on 6.0%, and the dosages of 5.0%, 6.0%, and 7.0% are used as three different dosages of waste vegetable oil regeneration-lubricating agent.

(2) Determination of regeneration time for waste vegetable oil

Through the comprehensive consideration of factors such as reading literature and actual construction, the regeneration times of waste vegetable oil were selected as 20 min, 40 min, and 60 min, and the time should not be too long.

### 2.2. Experimental Method of Microstructure

### 2.2.1. Performance Index Test of Aging Asphalt

According to the standard test methods of bitumen and bituminous mixtures for highway engineering in China, the test method is the same as that in Table 2. The test results of recycled aging asphalt with regeneration content are shown in Table 5.

**Table 5.** Physical performance index of recycled aging asphalt.

| Recycled Aging Asphalt | Penetration Degree (°C) | Softening Point (0.1 mm) | Ductility (mm) | Viscosity (Pa·s) |
|---|---|---|---|---|
| C# | 18.7 | 55.1 | 105 | $8.6 \times 10^{-1}$ |
| 5% + 20 min | 20.5 | 52.7 | 112 | $6.3 \times 10^{-1}$ |
| 5% + 40 min | 25.9 | 50.8 | 117 | $5.4 \times 10^{-1}$ |
| 5% + 60 min | 30.2 | 49.6 | 131 | $4.9 \times 10^{-1}$ |
| 6% + 20 min | 30.7 | 51.3 | 116 | $4.7 \times 10^{-1}$ |
| 6% + 40 min | 35.9 | 50.1 | 132 | $4.3 \times 10^{-1}$ |
| 6% + 60 min | 40.1 | 48.2 | 149 | $4.1 \times 10^{-1}$ |
| 7% + 20 min | 42.1 | 50.2 | 128 | $3.9 \times 10^{-1}$ |
| 7% + 40 min | 45.6 | 48.9 | 145 | $3.2 \times 10^{-1}$ |
| 7% + 60 min | 50.8 | 48.1 | 167 | $2.9 \times 10^{-1}$ |

### 2.2.2. Aging Asphalt Composition Experiment

Under different regeneration agent contents and different regeneration time conditions, a four-component experiment (according to the standard test methods of bitumen and bituminous mixtures for highway engineering in China, test method is T0618), the changes in each component before and after regenerating aging asphalt with waste vegetable oil are shown in Figure 1.

### 2.2.3. Infrared Spectrometer Experiment

In the test, the infrared spectrometer Nicolet IS 10 had the resolution set to 4 cm$^{-1}$, was scanned 32 times, and had the test wavelength range of 600–4000 cm$^{-1}$. The ranges of each group feature absorption peak are shown in Table 6.

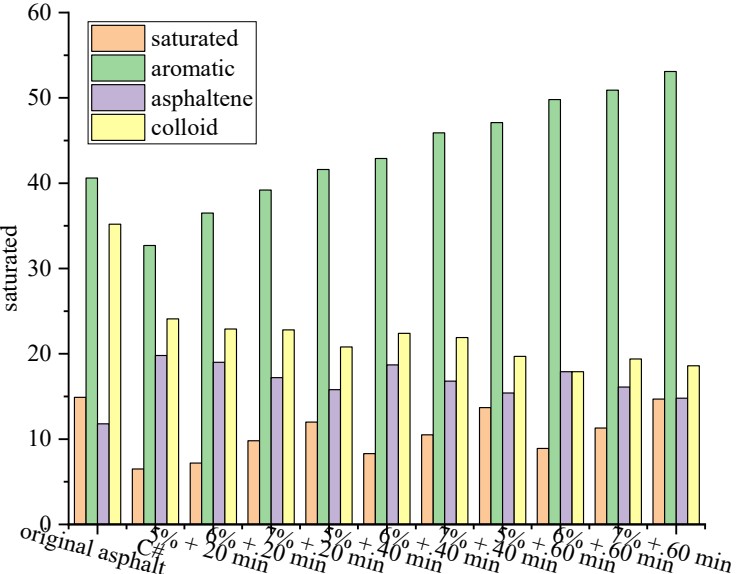

**Figure 1.** Changes in aging asphalt components.

**Table 6.** Absorption peak range of each group.

| Absorption Peak Range/cm$^{-1}$ | Main Group |
| --- | --- |
| 4000–3000 | O-H bond, N-H bond |
| 3300–2700 | C-H bond |
| 2500–1900 | Three-bond stretching vibrations of C-C, C-N |
| 1900–1500 | The stretching vibration of C-O double bond |
| 1500–1300 | The stretching vibration of aromatic rings |
| 1300–1000 | C-H bending vibration |
| 900–600 | Cis-trans configuration |

Regenerated aging asphalt, unregenerated aging asphalt (C#), and original asphalt were scanned by Fourier-transform infrared spectroscopy, and the infrared spectrum was obtained as shown in Figure 2. As the infrared spectra of the recycled asphalt for 5.0% and a regeneration time of 20 min were basically consistent with those of the original aging asphalt after infrared scanning, there is no infrared spectrum of the recycled asphalt for 5.0% + 20 min in Figure 2.

In the vicinity of 3050 cm$^{-1}$, the aging asphalt with the regeneration agent produced new, wide and short peaks, as shown in Figure 3.

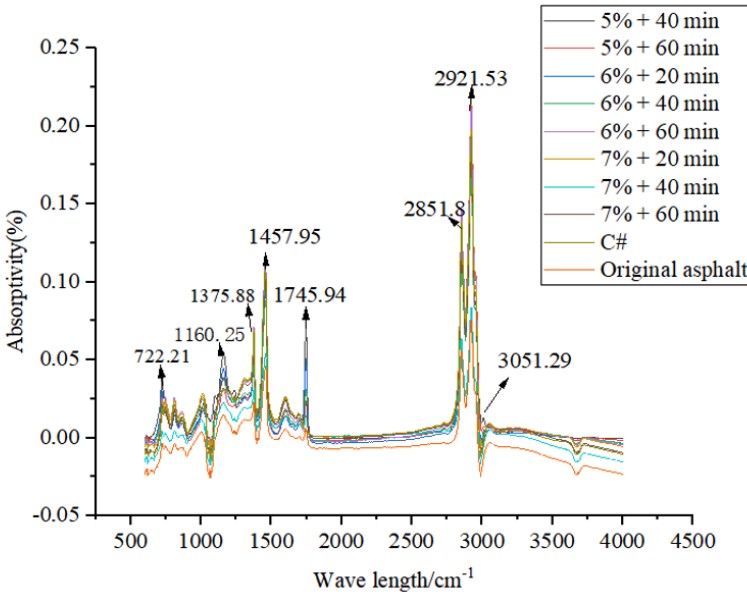

**Figure 2.** The infrared spectrum of each aging asphalt.

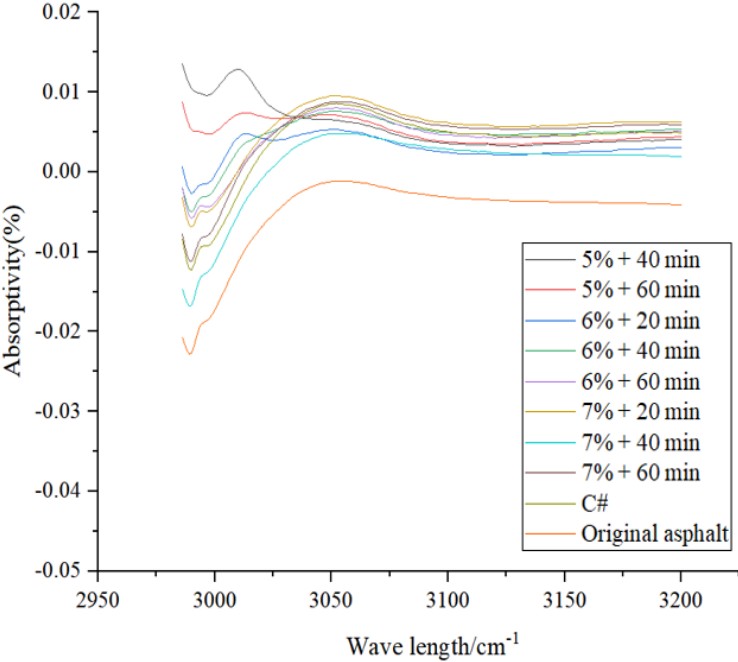

**Figure 3.** The infrared spectrum of asphalt around 3050 cm$^{-1}$.

The aging of asphalt led to the oxidation of a large number of benzyls and sulfides to form carbonyl functional groups and sulfoxide functional groups. Therefore, the carbonyl functional group index (CI) and sulfoxide functional group index (SI) were used to characterize the aging degree of asphalt. With the help of OMINC infrared data processing software, the CI value and SI value of aging asphalt under different regeneration agent contents and different regeneration times were obtained, and using calculation methods such as Formula (1) and Formula (2), the calculation results are shown in Figure 4.

$$CI = \frac{A_{C=O}}{A_{C-H}} \qquad (1)$$

$$SI = \frac{A_{S=O}}{A_{C-H}} \tag{2}$$

where:

$A_{C=O}$ is the hydroxyl characteristic peak area at 3050 cm$^{-1}$.

$A_{S=O}$ is the characteristic peak area of the sulfoxide group at 3050 cm$^{-1}$.

$A_{C-H}$ is the total peak area.

### 2.2.4. Atomic Force Microscope Observation Experiment

In order to observe the change in the apparent morphology of aging asphalt after regeneration, an atomic force microscope was used to observe the aging asphalt with different regeneration degrees. The magnification was 400 times and the test range was −25 nm–25 nm. The following figure shows the test results of the atomic force microscope before and after regeneration with waste vegetable oil.

(1)    Changes in the "honeycomb" structure

The formation of the "honeycomb" structure has a certain relationship with the content of each component in asphalt [20]. Wax crystal is the main structure of the honeycomb structure [21]. The change in the "honeycomb" structure of aging asphalt with the regeneration time and regeneration content is shown in Figure 5.

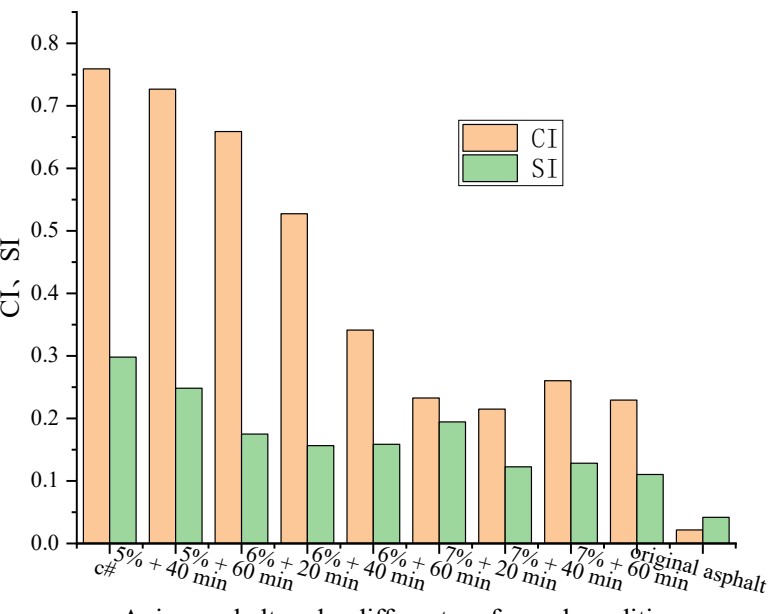

**Figure 4.** CI and SI values of aging asphalt after regeneration-moistening.

(2)    Microtopography of surface roughness

The apparent morphology of aging asphalt changes with the regeneration time and regeneration agent content, as shown in Figure 6.

The surface roughness of aging asphalt was calculated according to the calculation program in NanoScope Analysis software, and the surface roughness was quantified. The profile arithmetic average error (Ra) was used as the evaluation index. The calculation results are shown in Figure 7.

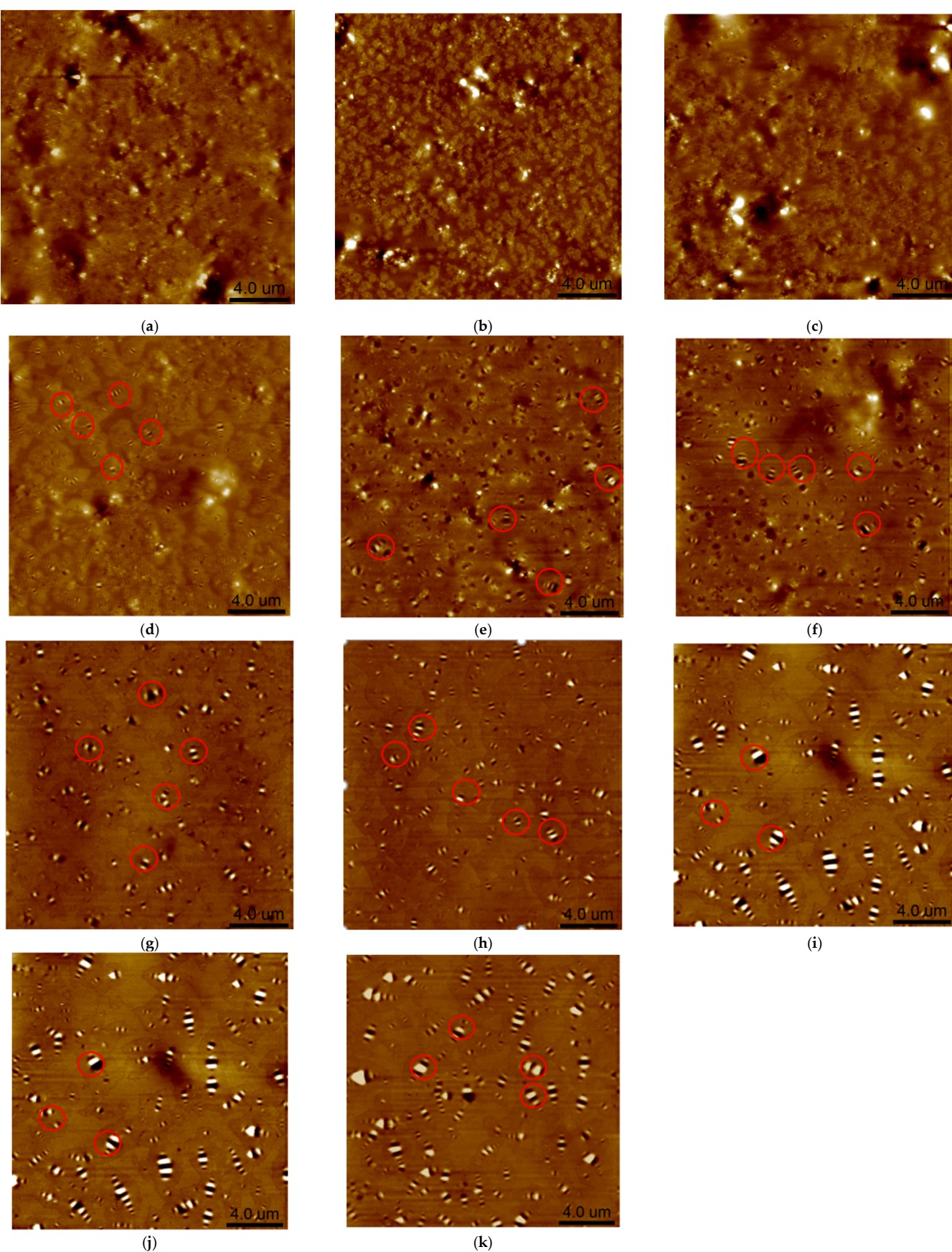

**Figure 5.** Changes in the "honeycomb" structure: (**a**) aging asphalt; (**b**) 5% + 20 min; (**c**) 5% + 40 min; (**d**) 5% + 60 min; (**e**) 6% + 20 min; (**f**) 6% + 40 min; (**g**) 6% + 60 min; (**h**) 7% + 20 min; (**i**) 7% + 40 min; (**j**) 7% + 60 min; and (**k**) original asphalt.

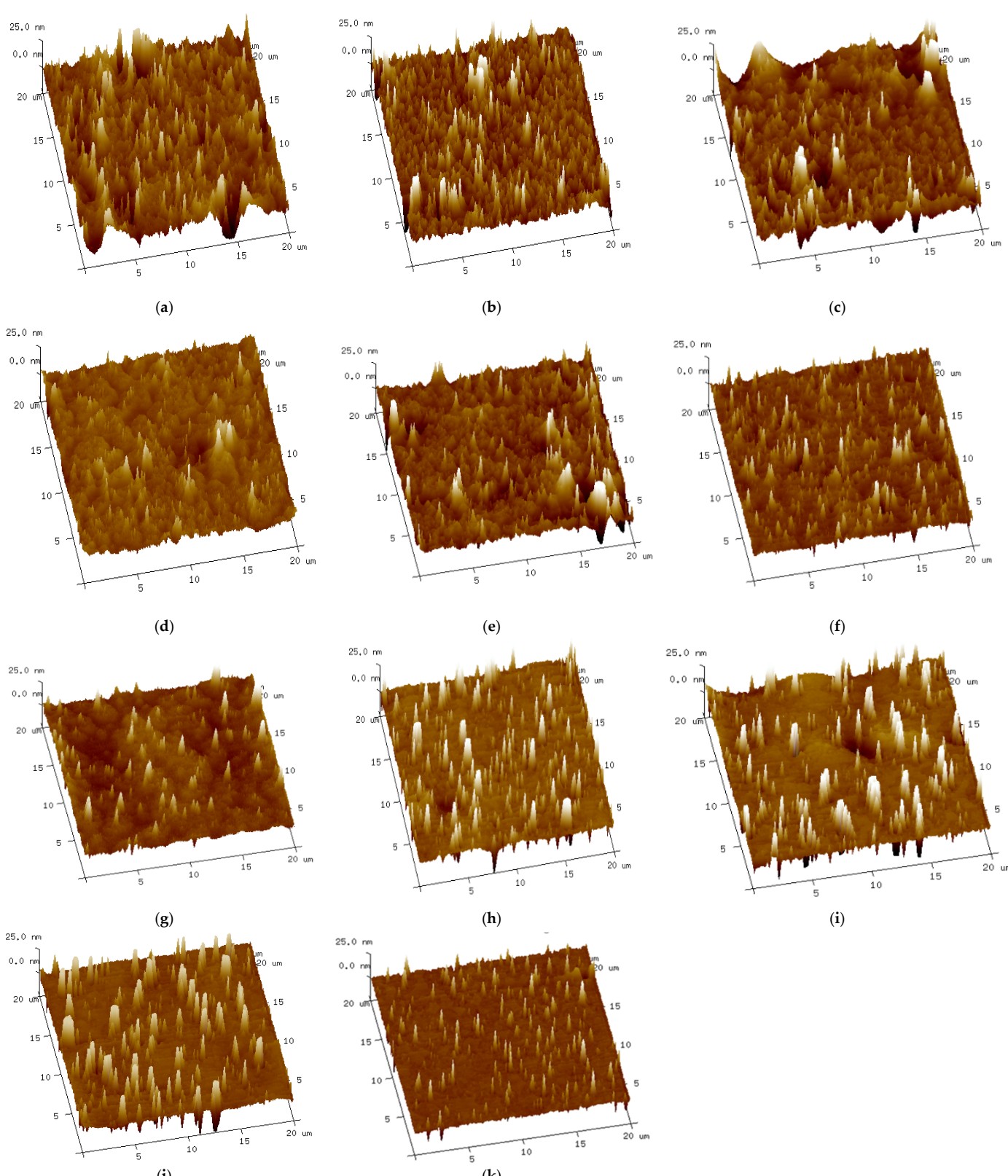

**Figure 6.** The 3D appearance of each aging asphalt: (**a**) aging asphalt; (**b**) 5% + 20 min; (**c**) 5% + 40 min; (**d**) 5% + 60 min; (**e**) 6% + 20 min; (**f**) 6% + 40 min; (**g**) 6% + 60 min; (**h**) 7% + 20 min; (**i**) 7% + 40 min; (**j**) 7% + 60 min; and (**k**) original asphalt.

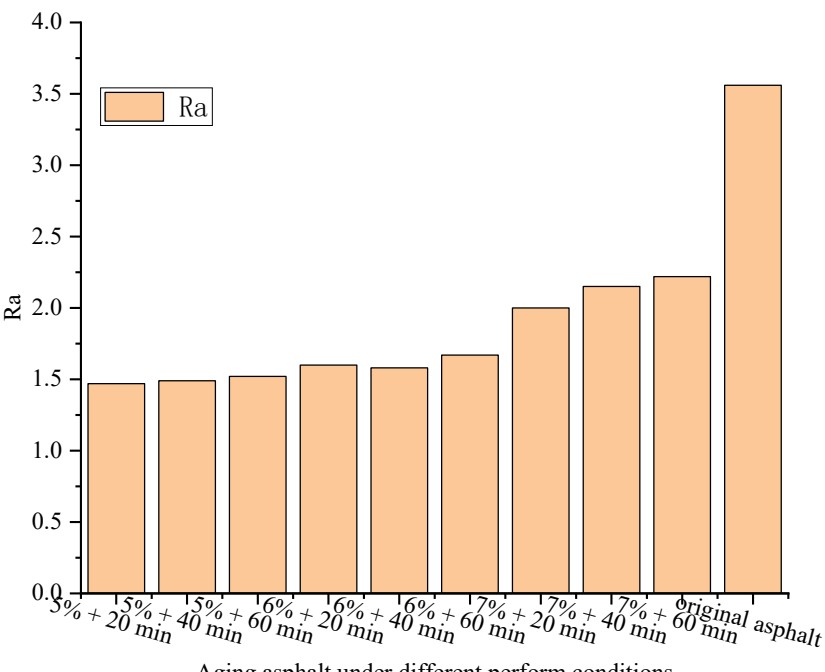

**Figure 7.** Ra value of contour arithmetic of aging asphalt after regeneration.

## 3. Results

### 3.1. Regeneration of Wasting Vegetable Oil

By comparing the test results of physical performance indexes such as the softening point and penetration of original asphalt and waste vegetable oil recycled asphalt in Tables 3 and 5, it can be found that after the regeneration effect of waste vegetable oil, the softening point and viscosity of aging asphalt decreased with the regeneration time and agent content, and the penetration and ductility increased with the regeneration time and agent content. When the waste vegetable oil content was 7.0% and the regeneration time was 60 min, the softening point at this time was restored to 80% of the original asphalt, the penetration was restored to about 75% of the original asphalt, and the ductility and viscosity were restored to more than 90% of the original asphalt. At this time, the physical performance was best.

### 3.2. Change in Aging Asphalt Composition

Under the same regeneration time, the saturated and aromatic contents increased linearly with the regeneration content, while the asphaltene and resin contents decreased with fluctuations. With the increases in the regeneration content and time, the saturated content of regenerated aging asphalt increased linearly as a whole, the aromatic content fluctuated, the asphaltene content decreased linearly, and the resin content decreased. When the dosage of the waste vegetable oil regeneration agent was 7.0% and the regeneration time was 60 min, the saturated and aromatic contents of regenerated aging asphalt were highest, which were 2.3 times and 1.6 times the original aging asphalt, respectively. The asphaltene and resin content of the regenerated aging asphalt decreased to varying degrees compared with the original aging asphalt, and the overall negative growth. When the dosage of the waste vegetable oil regeneration agent was 7.0% and the regeneration time was 60 min, the negative increase in asphaltene content was most obvious. When the content of the waste vegetable oil regeneration agent was 5.0% and the regeneration time was 60 min, the negative growth of resin content was most obvious, and the asphaltene and resin content of the original aging asphalt were 1.3 times those of the regenerated aging asphalt. Compared with the original asphalt, when the waste vegetable oil content was 7% and the regeneration time was 60 min, the saturated content of the recycled aged asphalt was 0.9 times that of the original asphalt, which was basically the same as that of

the original asphalt. The contents of aromatics and asphaltenes were 1.3 times those of the original asphalt, and the content of resin was 0.5 times that of the original asphalt.

During the aging process, the saturated and aromatic components of asphalt underwent aging reaction under the influence of high temperature, ultraviolet radiation, and other factors, which was converted into resin and then into asphaltene. Therefore, the asphaltene and resin contents of the regenerated aging asphalt were higher than those of the original asphalt. In the figure, the variation in asphaltene and resin showed negative growth. The content of asphaltene and resin decreased with the regeneration time and agent dosage, and the decrease in the two components was equivalent, with a change rate of 10–20%. Under the same amount of regeneration agent, when the regeneration time was 60 min, the content of the two-hour component of asphalt reached the minimum.

### 3.3. Test Results of Infrared Spectrometer

The peaks in the figures are common portions, of which the peaks of 3051.29 cm$^{-1}$ represent the presence of the aromatic ring. The peaks of 2921.99 cm$^{-1}$ and 2852.88 cm$^{-1}$ represent the peaks caused by the vibration of the alphabetic group; 1745.94 cm$^{-1}$ represents the peak caused by the vibration of the hydroxy functional group carbon–oxygen bond; according to the predecessors, the absorption peak of waste vegetable oil A is attributed to the carbonyl C=O in carbonate. The peak at 1457.96 cm$^{-1}$ represents a saturated C=H key feature of the amide substance.

The peaks in the figure are the common part, in which the peak near 3050 cm$^{-1}$ represents the existence of aromatic rings. At the dosage of 7.0% waste vegetable oil, the peak is relatively high, and the area of the absorption peaks decreases with the regeneration time. The peaks near 2920 cm$^{-1}$ and 2850 cm$^{-1}$ represent the vibrations of methylene symmetric and antisymmetric functional groups, and the peaks near 1745 cm$^{-1}$ represent the vibrations of hydroxyl functional groups. Waste vegetable oil has the absorption peak of carbonyl C=O in carbonyl acid. Representative amides near 1460 cm$^{-1}$ can be characterized by a saturated C=H bond. The vicinity of 1375 cm$^{-1}$ represents the characteristic peak caused by methyl ($-CH_3$) vibration. The peak near 1160 cm$^{-1}$ represents the stretching vibration of C-F on the fluorine atom and benzene ring. The vicinity of 720 cm$^{-1}$ represents the characteristic peak caused by the vibration of the methylene chain.

The characteristic peak near the wavelength of 3050 cm$^{-1}$ indicates that there are aromatic rings. The peak is relatively high at a 7.0% waste vegetable oil content, and the area of the absorption peak decreases with the regeneration time.

With the increase in regeneration agent dosage and regeneration time, the CI value and SI value decreased. When the dosage of the waste vegetable oil regeneration agent reached 7.0% and the regeneration time was 40 min, the CI value and SI value showed an upward trend, and then with the increase in regeneration time, the CI value and SI value of aging asphalt decreased gradually, and the overall trend declined. When the dosage of the waste vegetable oil regeneration agent was 7.0% and the regeneration time was 20 min, the CI value and SI value of aging asphalt after regeneration reached the lowest value, which decreased by 71% and 58%, respectively, compared with aging asphalt. At this time, the CI value was 10.6 times that of the original asphalt, and the SI value was 2.6 times that of the original asphalt. Overall, the CI and SI values of aging asphalt with a waste vegetable oil content of 7% and regeneration time of 60 min were the closest to those of the original asphalt. This phenomenon indicated that the addition of waste vegetable oil inhibited the oxidation of benzyl and sulfide, weakened the interaction between polar functional groups in aging asphalt, and reduced the contents of carbonyl functional groups and sulfoxide functional groups. With the increase in the CI value and SI value, the aging degree of asphalt gradually increased.

### 3.4. Atomic Force Microscope Test Results

It can be seen from the red circle in Figure 5 that the honeycomb structure of aging asphalt was basically not shown in (a). When the content of the waste vegetable oil regeneration agent was 5.0% and the regeneration time was 60 min, a few honeycomb structures could be observed in (d). In (e), (f) and (g), 6.0% of the waste vegetable oil regeneration agent content compared to the 5.0% honeycomb structure had a more concentrated distribution. When the dosage of the regeneration agent was 7.0%, it can be seen that the honeycomb structure was more obvious and the number significantly increased, and the distribution was more uniform. No honeycomb structure was found in aging asphalt without the waste vegetable oil regeneration agent. When the regeneration time was 60 min and the dosage of the waste vegetable oil regeneration agent was 5.0%, a few honeycomb structures were found and the honeycomb structure was not obvious (see the red ring outer section). With the increase in regeneration time and agent content, the honeycomb structure increased significantly. When the regeneration content of the waste vegetable oil was 7.0%, the honeycomb structure changed little with the increase in regeneration time, and the distribution range gradually expanded to be uniform.

The apparent 3D morphology diagram corresponds to the bee structure diagram. When the bee structure gradually increased, the peak number of asphalt surface bulges in the apparent 3D morphology diagram increased. When the dosage of the regeneration agent was 7.0%, the honeycomb structure was larger and the distribution was more standardized. Through (h), (i) and (j), it can be seen that the convex position distribution of each peak on the surface of aging asphalt was more uniform, which further shows that the recovery of the honeycomb structure had a direct impact on the apparent morphology of asphalt.

According to the calculation results above, the surface roughness of aging asphalt without waste vegetable oil was smallest. With the addition of the waste vegetable oil, the roughness of the asphalt surface increased. When the dosage of the waste vegetable oil regeneration agent was 7.0%, the Ra value increased with the regeneration time and reached the maximum value at 60 min. Compared with the Ra value of the original asphalt, the difference between the Ra value of waste vegetable oil and original asphalt was smallest when the content of waste vegetable oil was 7% and the regeneration time was 60 min.

Based on the above analysis, the optimal regeneration time of waste vegetable oil was 60 min, and the optimal regeneration content was 7.0%. At this time, the physical properties, microstructure analysis, and microstructure were well restored.

### 3.5. Correlation of Microstructure Characteristics of Aging Asphalt

The variations in component content in the aging process of the regenerated aging asphalt with different regeneration times and contents, carbonyl functional group indexes, and sulfoxide functional group indexes that characterize the aging degree of asphalt, and the average deviation of the contour arithmetic of roughness that characterizes the micro-morphology in the aging process were used to establish the models, respectively, by MATLAB software. The fitting effect of each model is shown in Figure 8.

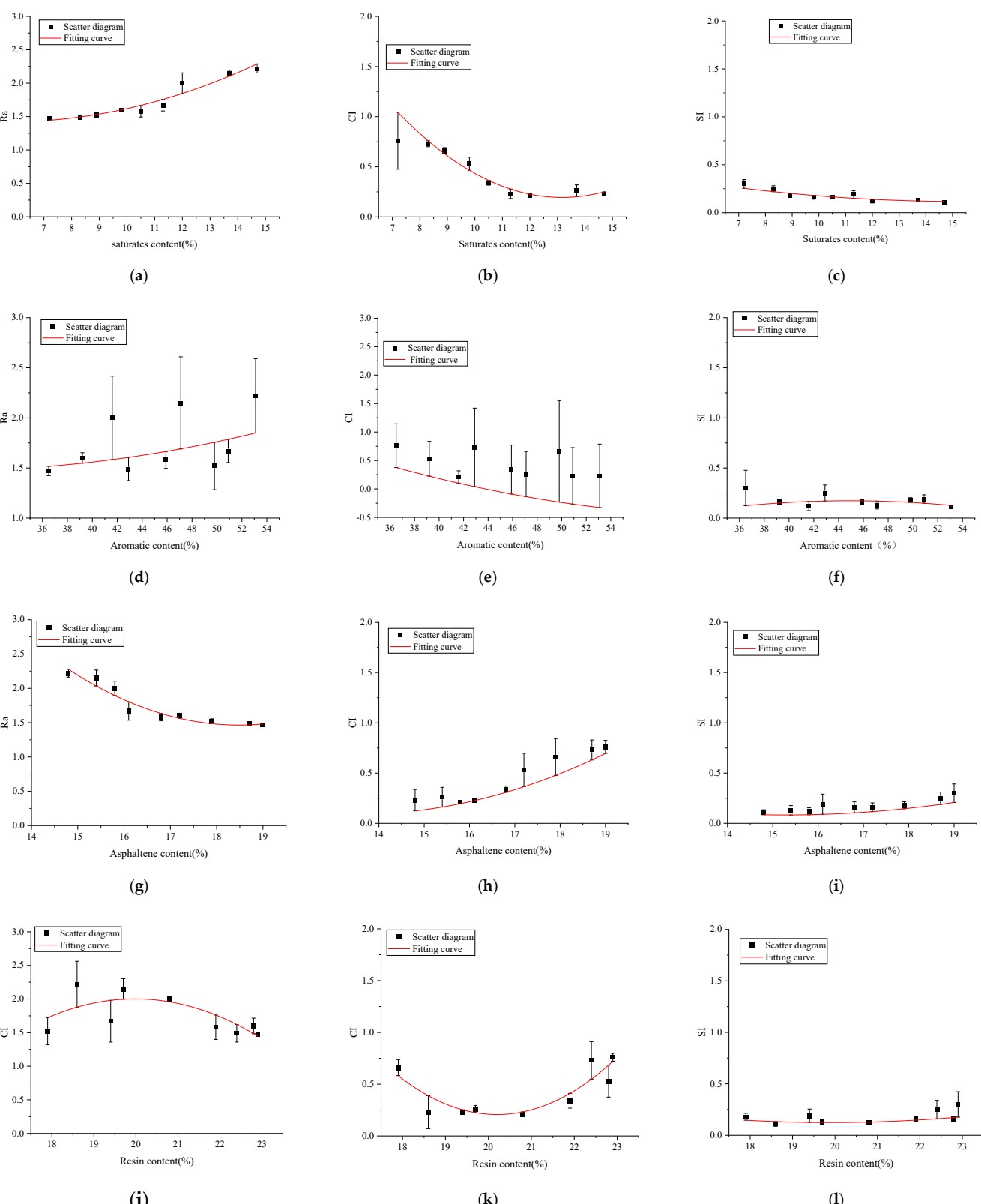

**Figure 8.** Fitting diagram of regenerated aging asphalt components and microscopic characteristic model: (**a**) Fitting effect of saturation content and Ra; (**b**) fitting effect of saturation content and CI; (**c**) fitting effect of saturation content and SI; (**d**) fitting effect of aromatic content and Ra; (**e**) fitting effect of aromatic content and CI; (**f**) fitting effect of aromatic content and SI; (**g**) fitting effect of asphaltene content and Ra; (**h**) fitting effect of asphaltene content and CI; (**i**) fitting effect of asphaltene content and SI; (**j**) fitting effect of resin content and Ra; (**k**) fitting effect of resin content and CI; (**l**) fitting effect of resin content and SI.

Optimal models of component content and microscopic characteristics of the waste vegetable oil-regenerated aging asphalt were established. Among them, the fitting effect of saturates content of the regenerated aging asphalt and CI representing the aging degree of asphalt was best, and the correlation coefficient reached 0.959. The correlation between the saturates content, asphaltene content, and their quantitative index of microscopic morphology characteristics was above 0.5, indicating that the change in saturates content and asphaltene content had the greatest correlation with the microscopic morphology characteristics. The fitting effect of aromatic content of the regenerated aging asphalt with SI was worst, and the correlation coefficient was only 0.114. The correlation with other microscopic morphology characteristics was all below 0.5, indicating that the change in aromatic content had little correlation with its microscopic morphology characteristics. The fitting effect of regenerated aging asphalt resin content with CI was relatively good, with a correlation coefficient of 0.697, and the fitting effect with SI was poor, with a correlation coefficient of only 0.203, indicating that the change in resin content had a good correlation with the carbonyl functional group index, and had little correlation with sulfoxide functional group index. The correlation between the change in resin content and the overall characteristics of micro-morphology was not ideal.

## 4. Discussion

After aging, the light components of the original asphalt were seriously dispersed and the component content was unbalanced. Saturates were dispersed, and aromatics were aromatic hydrocarbons, which are similar to the oil in asphalt and jointly determine the stability of asphalt colloids. Gum and asphaltene contain similar elements, which jointly determine the viscosity performance of asphalt. Saturates and aromatics are volatile at high temperatures. After the waste vegetable oil regeneration agent was added, the missing components of aging asphalt were supplemented, and the saturated and aromatic contents of the aging original asphalt were increased. At the same time, the conversion process of the two into colloid and asphaltene was inhibited.

Through the analysis of the influence of the waste vegetable oil regeneration agent on the functional groups of aging asphalt, the new vibration characteristic peaks of each aging asphalt after the full regeneration of the regeneration agent were lower, which further indicates that the waste vegetable oil regeneration agent could not only fully integrate with the aging asphalt, but also the chemical performance of the aging asphalt did not change. The addition of the regeneration agent weakened the interaction between polar functional groups in aging asphalt, reduced the content of carbonyl functional groups and sulfoxide functional groups, and inhibited the oxidation of benzyl and sulfide, thereby slowing down the aging degree of asphalt.

Through the analysis of the change in the micro-morphology of aging asphalt by the waste vegetable oil regeneration agent, with the increase in the dosage and regeneration time of the waste vegetable oil regeneration agent, the honeycomb structure started from scratch, the number increased, and the morphology changed from the concentrated distribution to the uniform distribution, which can explain why the regeneration of the waste vegetable oil regeneration agent had a certain regeneration effect and could restore some structures.

Optimal models for each component of the regenerated aging asphalt and its micro-morphology characteristics were established, which are quadratic and exponential models, respectively. Through the analysis of the model fitting effect, under the conditions of different regeneration contents and times, the changes in saturated content and asphaltene content had the greatest correlation with its micro-morphology characteristics, and the changes in aromatic content had the smallest correlation with its micro-morphology characteristics.

## 5. Conclusions

The regeneration of aged asphalt in RAP with waste vegetable oil as a regeneration agent can promote the reuse of waste vegetable oil. Different waste vegetable oil contents and regeneration times affect the regeneration degree and performance of aged asphalt. Through the analysis of the results of functional groups, the addition of waste vegetable oil did not cause the chemical changes of aged asphalt. When the waste vegetable oil regeneration content was 7% and the regeneration time was 60 min, the ductility and viscosity of the recycled aged asphalt proposed in this study were restored to more than 90% of the original asphalt, and the softening point and penetration were restored to more than 70% of the original asphalt. The contents of saturated and aromatic components were twice those of aging asphalt. The honeycomb structure was evenly distributed and the largest number. The quantitative index of surface roughness reached a large value, and the regeneration effect of aged asphalt was best. Waste vegetable oil can be used as a regeneration agent to realize the regeneration of aged asphalt at room temperature, which is the test condition and provides for the research of asphalt cold recycling mixtures in the future. It is expected to improve the defects of road performance of the traditional asphalt cold recycling mixture.

**Author Contributions:** Z.S.: Writing—Original Draft Preparation, Experiments—Design, Accomplishment and Analysis, Writing—Review and Editing, Review and Supervision. L.N.: Writing—Original Draft Preparation, Experiments—Design, Accomplishment and Analysis, Writing—Review and Editing, F.X.: Experiments—Design, Accomplishment and Analysis, Writing—Review and Editing. X.B.: Review and Supervision. All authors have read and agreed to the published version of the manuscript.

**Funding:** This research was funded by the National Natural Science Foundation of China under the project numbers Z20099 and ZH17032.

**Institutional Review Board Statement:** Not applicable.

**Informed Consent Statement:** Informed consent was obtained from all subjects involved in the study.

**Data Availability Statement:** MDPI journals are members of the Committee on Publication Ethics (COPE). We fully adhere to its Code of Conduct and to its Best Practice Guidelines. The data presented in this study are available on request from the corresponding author.

**Conflicts of Interest:** The authors declare no conflict of interest.

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
