# Peer review of "The Effect of Waste Plant Oil on the Composition and Micro-Morphological Properties of Old Asphalt Composition"

_buildings, doi:10.3390/buildings11090407_

Round 1

Reviewer 1 Report

The major problem encountered in reading this article is the total lack of analysis carried out on the reference sample (virgin bitumen). Therefore, in my opinion, it is not possible to prove the validity of a rejuvenator agent if the results obtained on aged samples are not compared with those obtained on the unaged sample.

Whitin the article the term preformed is often used both as preformed time and as preformed agent. It would be helpful explain what is meant. Furthermore, how were the four components shown in figure 1 obtained?

In paragraph 2.1.3 line 103-104, it is not clear whether the aged asphalt C# is aged again, since its aging had alredy been described in line 97.

In paragraph 2.1.1, the authors talk about an initial phase of the research, referring to two articles 22 and 23, unfortunately the authors of the two cited articles are not the same.

Please standardize the bibliography. Sometimes the articles are cited with the names of all the authors, other times only the first name is put and then et al.   

Reviewer 2 Report

This manuscript reports an experimental study on using waste vegetable oil as the main component of the regeneration agent to prepare preformed aging asphalt. Sustainable development of the infrastructures has become an urgent matter to cope with the climate change, and recycling of waste matters as alternative construction materials has been extensively investigated in the past few years among which include the attempt of using waste cooking oil as regeneration agent of aged bitumen binder. The current research is within the content of such efforts, and therefore the involved topic is interesting. This research aims to exploit the micromorphological characteristics of the virgin and recycled bitumen binders. The text of this manuscript is clear and easy to follow, and the major conclusions were well supported by the experimental results. The findings from this manuscript contribute to the mechanistic understanding of the modification mechanism of waste oil on bitumen binder. It is suggested to be published in the journal of Buildings before the following comments are considered.

  1. Please carefully check the English writing of the current manuscript. There exists too many grammar errors.
  2. The structure of this study must be adjusted. For instance, the test results included in section 2 should be presented in the section 3.
  3. The format of all the figures should be unified.

Round 2

Reviewer 1 Report

Unfortunately, in my opinion, the physical analysis alone (ductility, softening point, etc etc) are not sufficient to test the validity of regeneration agent. In my opinion in would be useful to carrie out measures of IR, AFM and determination of satures, aromatics, asphaltene and colloid also on virgin bitumen. This is in order to evaluate the effect of oil on bitumen.

Finally, in the reply to previous comments (Point 2) the authors used the term "perform", but the term "preform" (preformed aging, preformed time) continous to be present in the text. Please, make the text uniform.

In table 1, please write the ductility with scientific notation.
